# BMP2 and BMP7 cooperate with H3.3K27M to promote quiescence and invasiveness in pediatric diffuse midline gliomas

Paul Huchede[1†], Swann Meyer[1†], Clément Berthelot[1†], Maud Hamadou[1†], Adrien Bertrand-Chapel[1], Andria Rakotomalala[2], Line Manceau[3], Julia Tomine[1], Nicolas Lespinasse[1], Paul Lewandowski[2], Martine Cordier-Bussat[1], Laura Broutier[1], Aurélie Dutour[1], Isabelle Rochet[4], Jean-Yves Blay[1], Cyril Degletagne[5], Valéry Attignon[5], Angel Montero-Carcaboso[6], Marion Le Grand[7], Eddy Pasquier[7], Alexandre Vasiljevic[4], Pascale Gilardi-Hebenstreit[3], Samuel Meignan[2], Pierre Leblond[1,8], Vanessa Ribes[3‡], Erika Cosset[9‡], Marie Castets[1*‡]

[1]Childhood Cancer & Cell Death (C3) team, LabEx DEVweCAN, Institut Convergence Plascan, Centre Léon Bérard, Centre de Recherche en Cancérologie de Lyon (CRCL), Université Claude Bernard Lyon 1, INSERM 1052, CNRS 5286, Lyon, France; [2]University of Lille, CNRS, Inserm, CHU Lille, UMR9020-U1277-CANTHER Cancer Heterogeneity Plasticity and Resistance to Therapies, Centre Oscar Lambret, Lille, France; [3]Université Paris Cité, CNRS, Institut Jacques Monod, Paris, France; [4]Multisite Institute of Pathology, Groupement Hospitalier Est du CHU de Lyon, Hôpital Femme-Mère Enfant, Bron, France; [5]Platform of Cancer Genomics, Centre Léon Bérard, Lyon, France; [6]Preclinical Therapeutics and Drug Delivery Research Program, Department of Oncology, Hospital Sant Joan de Déu, Barcelona, Spain; [7]Centre de Recherche en Cancérologie de Marseille (CRCM), Université Aix-Marseille, Institut Paoli- Calmettes, Centre de Lutte Contre le Cancer de la région PACA, INSERM 1068, CNRS 7258, Marseille, France; [8]Department of Pediatric Oncology, Institute of Pediatric Hematology and Oncology (IHOPe), Centre Léon Bérard, Lyon, France; [9]GLIMMER Of lIght (GLIoblastoma MetabolisM, HetERogeneity, and Organoids) team, Centre Léon Bérard, Centre de Recherche en Cancérologie de Lyon (CRCL), Université Claude Bernard Lyon 1, INSERM 1052, CNRS 5286, Lyon, France

*For correspondence:
marie.castets@lyon.unicancer.fr

†These authors contributed equally to this work

‡Co-senior author

Competing interest: The authors declare that no competing interests exist.

**Abstract** Pediatric diffuse midline gliomas (pDMG) are an aggressive type of childhood cancer with a fatal outcome. Their major epigenetic determinism has become clear, notably with the identification of K27M mutations in histone H3. However, the synergistic oncogenic mechanisms that induce and maintain tumor cell phenotype have yet to be deciphered. In 20 to 30% of cases, these tumors have an altered BMP signaling pathway with an oncogenic mutation on the BMP type I receptor ALK2, encoded by *ACVR1*. However, the potential impact of the BMP pathway in tumors non-mutated for *ACVR1* is less clear. By integrating bulk, single-cell, and spatial transcriptomic data, we show here that the BMP signaling pathway is activated at similar levels between *ACVR1* wild-type and mutant tumors and identify BMP2 and BMP7 as putative activators of the pathway in a specific subpopulation of cells. By using both pediatric isogenic glioma lines genetically modified to overexpress H3.3K27M and patients-derived DIPG cell lines, we demonstrate that BMP2/7 synergizes with H3.3K27M to induce a transcriptomic rewiring associated with a quiescent but invasive cell state. These data suggest a generic oncogenic role for the BMP pathway in gliomagenesis of

pDMG and pave the way for specific targeting of downstream effectors mediating the K27M/BMP crosstalk.

## eLife assessment

This **valuable** study examines whether the BMP signaling pathway has a role in H3.3K27M DMG tumors, regardless of the presence of ACRVR1 activating mutations. The authors provide **solid** evidence that BMP2/7 synergizes with H3.3K27M to induce a transcriptomic rewiring associated with a quiescent but invasive cell state. Although this work could be further enhanced by the inclusion of additional models, the study overall points to BMP2/7 as a potential target for future therapies in this deadly cancer.

## Introduction

pDMG, including Diffuse Intrinsic Pontine Gliomas (DIPG), are rare and aggressive brain tumors that arise in the pons, thalamus, or spinal cord of children, most commonly between the ages of 5 and 10 (*Mackay et al., 2017*; *Di Ruscio et al., 2022*; *Sulman and Eisenstat, 2021*). pDMG are almost uniformly fatal, with a median overall survival of 9–11 months (*Khuong-Quang et al., 2012*; *Hoffman et al., 2018*), thereby representing the leading cause of mortality in pediatric neuro-oncology (*Mackay et al., 2017*; *Sulman and Eisenstat, 2021*). Clinical management of pDMG and especially of DIPG is a major challenge given their location in vital nervous centers and their leptomeningeal dissemination, which prevent any prospect of surgical intervention (*Sethi et al., 2011*). Radiotherapy, the current standard of care, is at best only transiently effective (*Sulman and Eisenstat, 2021*; *Vitanza and Monje, 2019*). Moreover, pDMG are highly resistant to currently available chemotherapies although promising combinations of molecules, including ONC201, are currently in clinical trials (*Jackson et al., 2023*).

High-throughput sequencing demonstrated that pDMG is associated with a major disruption of the epigenetic landscape, resulting in 80% of cases from a lysine-to-methionine substitution at position 27 (K27M) in genes encoding histone variants H3.3 (*H3F3A/H3-3A*) or H3.1 (*HIST1H3B/H3C2* or *HIST1H3C/H3C3*) (*Khuong-Quang et al., 2012*; *Schwartzentruber et al., 2012*; *Sturm et al., 2012*; *Wu et al., 2012*). H3K27M variants, also known as oncohistones, are affine for EZH2, the methyltransferase subunit of Polycomb Repressive Complex 2 (PRC2) and inhibit its activity. As a result, pDMG show abnormal methyl group deposition on H3K27 and a global loss of trimethylated H3K27 (H3K27me3) (*Bender et al., 2013*; *Chan et al., 2013*; *Lewis et al., 2013*; *Piunti et al., 2017*; *Stafford et al., 2018*; *Harutyunyan et al., 2019*), a histone mark associated with transcriptional repression.

In addition to *TP53*, other genetic alterations have been described, notably in the *PDGFR*, *EGFR,* or *PIK3CA* genes (*Mackay et al., 2017*), suggesting an oncogenic synergy between H3K27-based epigenetic remodeling and the activation of several transcriptional programs (*Nagaraja et al., 2019*; *Huchedé et al., 2022*). Accordingly, 20 to 30% of pDMG cases are associated with mutations in the *ACVR1* gene, encoding for the bone morphogenetic protein (BMP) type I receptor ALK2 (*Buczkowicz et al., 2014*; *Fontebasso et al., 2014*; *Taylor et al., 2014*; *Wu et al., 2014*), which leads to the overactivation of intracellular BMP signaling pathway (*Carvalho et al., 2019*; *Fortin et al., 2020*; *Haupt et al., 2018*; *Hoeman et al., 2019*; *Ramachandran et al., 2021*; *Jessa et al., 2022*; *Mucha et al., 2018*; *Hatsell et al., 2015*; *Hino et al., 2015*). Such increase in BMP signaling in the pDMG epigenetic context has been suggested to promote tumorigenesis by maintaining cells in a proliferative, mesenchymal-like, and undifferentiated state in vitro and in vivo (*Carvalho et al., 2019*; *Fortin et al., 2020*; *Hoeman et al., 2019*; *Jessa et al., 2022*). In 69% of cases, these *ACVR1* mutations are associated with the H3.1K27M mutation, whereas less than 20% are observed in H3.3K27M tumors (*Mackay et al., 2017*). The question is then whether the BMP signaling pathway could also be involved in the tumorigenesis of H3.3K27M-positive tumors. Recent data support the idea that H3.1K27M tumors mutated for *ACVR1* would emerge from a ventral pool of oligodendrocyte progenitor cells (OPC) characterized by the expression of the transcription factor NKX6-1 and dependent on the Sonic Hedgehog (SHH) signaling (*Jessa et al., 2022*). In contrast, H3.3K27M is thought to preferentially derive from dorsal progenitors expressing the dorsal PAX3- and BMP-dependent transcription factor (*Jessa et al., 2022*). In other words, H3.1K27M tumors would depend on the concomitant acquisition

of the *ACVR1* mutation for transformation, whereas the BMP pathway could be activated in a tumor-independent manner in H3.3K27M tumors, due to their BMP-rich microenvironment. However, while the pro-tumorigenic activity of BMP signaling is clear in *ACVR1* mutant pDMG, recent data have unexpectedly reported that BMP ligands may exert a tumor-suppressive activity in H3.3K27M-*ACVR1* wild-type (WT) pDMG cellular models (*Sun et al., 2022*).

Here, we have integrated bulk, single-cell and spatial transcriptomic data from patients with functional approaches in cellular models to characterize the impact of BMP activation in H3.3K27M-*ACVR1* WT pDMG. Bioinformatic analyses indicate that BMP signaling pathway activation ground state is independent of *ACVR1* status in pDMG, which likely results both from BMP2/7 tumor-autonomous and microenvironment-driven signals in *ACVR1* WT tumors. Functional modeling on pediatric glioma cell lines and spatial transcriptomics unveil that H3.3K27M and BMP2/7 synergize to induce a transcriptomic switch leading to a quiescent but invasive cell state. These results shed new light on the complex phenotype resulting from the synergy between activation of the BMP pathway and epigenetic remodeling induced by the H3.3K27M mutation in pDMG, and the interest of a therapeutic approach targeting the downstream oncogenic pathways responsible for the invasive potential of tumor cells.

## Results

### BMP pathway activation level is independent of *ACVR1* mutational status in pDMG, and likely supported by BMP2/BMP7 in H3.3K27M tumors

Oncogenic (*Carvalho et al., 2019*; *Fortin et al., 2020*; *Hoeman et al., 2019*; *Jessa et al., 2022*) and tumor-suppressive (*Sun et al., 2022*) functions of the BMP pathway have both been reported in pDMG. To clarify this point, we first compared tumor clustering according to their *ACVR1* status, based on the transcriptional profiling of 193 BMP pathway target genes as a reliable readout of the pathway activation state, including notably the downstream transducers *ID1*, *ID2*, *RUNX2*, and *GATA3* (see list in *Supplementary file 1a*). Principal component analysis (PCA) performed on this BMP target genes subset demonstrates no segregation of *ACVR1* mutant (in blue) from *ACVR1* WT samples (in red) in two independent publicly available transcriptomic cohorts of 31 (*McLeod et al., 2021*) and 40 (*Mackay et al., 2017*) patients (*Figure 1A* and *Figure 1—figure supplement 1A*, respectively). Consistently, the expression pattern of these genes does not segregate pDMG according to their *ACVR1* status in an unsupervised hierarchical clustering analysis in both cohorts (*Figure 1B* and *Figure 1—figure supplement 1B*). To have a more comprehensive overview of BMP pathway activation level, we used the PROGENy method to infer downstream TGF-β/BMP response footprint from perturbation-response genes indicative of the pathway activity (*Schubert et al., 2018*). Interestingly, we observed no significant difference in the TGF-β/BMP activation PROGENy score between *ACVR1* mutant and WT tumors (*Figure 1C* and *Figure 1—figure supplement 1C*). Of note, although variable, this score was even higher than the maximum observed in *ACVR1* mutant tumors in 12.5 (cohort 2)–18.5% (cohort 1) of *ACVR1* WT pDMG.

To define what alternative mechanism could lead to the activation of the BMP pathway in *ACVR1* WT tumors, we profiled the expression of all major BMP pathway components. Compatible with BMP signaling being active, several ligands, receptors, antagonists and intracellular transducers are expressed in *ACVR1* WT pDMG, albeit at expression levels very similar to those observed in *ACVR1* mutant tumors (*Figure 1B and D* and *Figure 1—figure supplement 1B, D*). By differential analysis, we found that only *CHRDL1* is significantly overexpressed in *ACVR1* WT tumors compared to mutant ones in two independent transcriptomic cohorts, as previously reported in pDMG cell lines (*Sun et al., 2022*; *Figure 1D* and *Figure 1—figure supplement 1D*). Although its role in regulating the BMP pathway remains to be fully elucidated (*Nakayama et al., 2001*; *Sakuta et al., 2001*; *Larman et al., 2009*; *Blanco-Suarez et al., 2018*; *Lin et al., 2005*), the gain of *CHRDL1* expression is unlikely promoting the activation of the BMP pathway in *ACVR1* WT tumors. Since no difference is observed in receptor expression consistently between the two cohorts (*Figure 1D* and *Figure 1—figure supplement 1D*), we hypothesized that activation of this pathway likely results from tumor-autonomous and/or microenvironment-driven production of ligands. Out of all BMP ligands present in pDMG tumors, BMP2 and BMP7 are the two most highly expressed ligands in H3.3K27M pDMG in both cohorts with no significant difference according to H3 and ACVR1 mutational status (*Figure 1E*

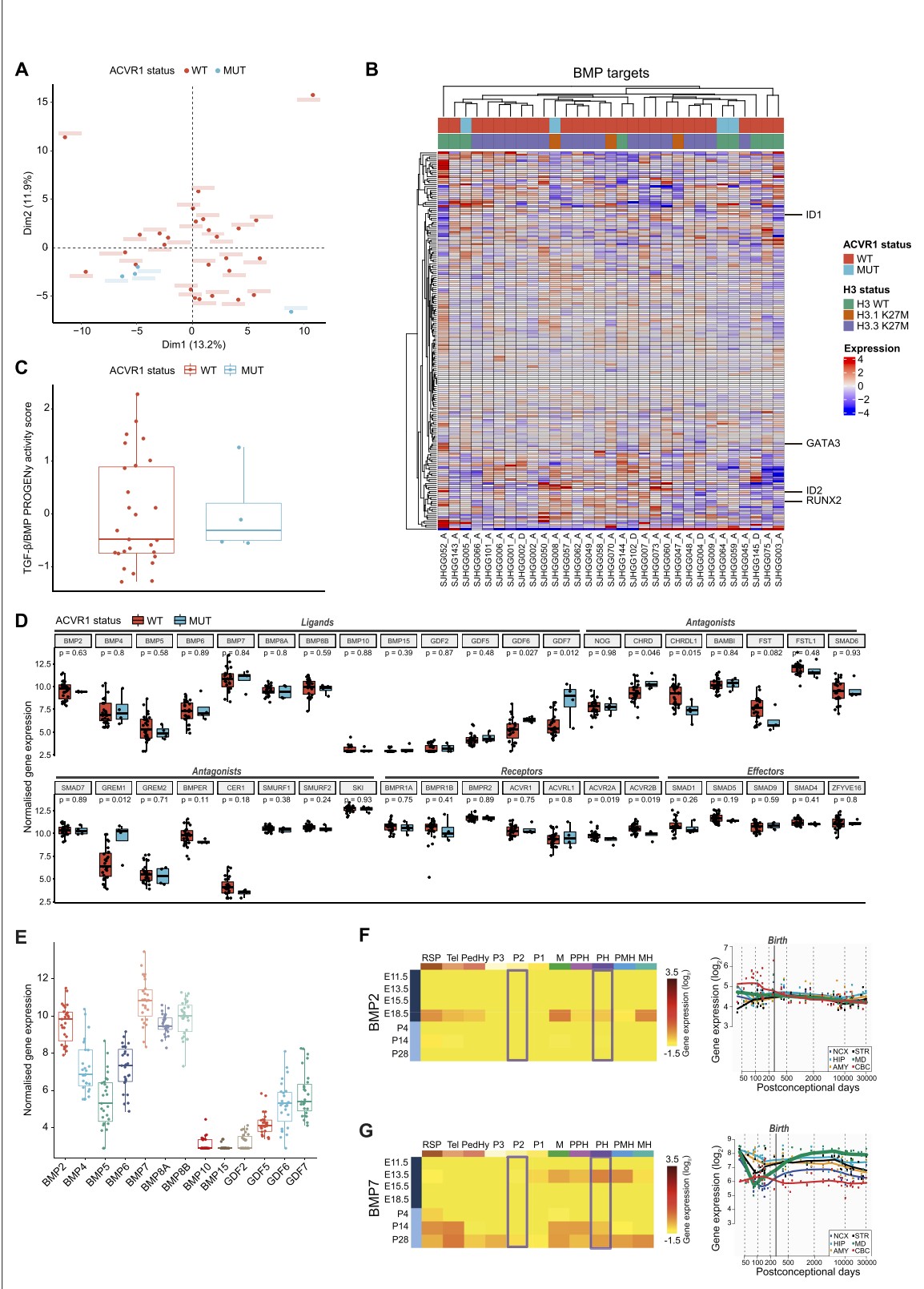

**Figure 1.** Bone morphogenetic protein (BMP) pathway activation coincides with high BMP2/7 levels in *ACVR1* WT-H3.3K27M pDMG. (**A**) Principal component analysis (PCA) of *ACVR1* WT (red) and mutant pediatric diffuse midline gliomas (pDMG) (blue) samples based on transcriptomic data from cohort 1. (**B**) Heatmap representing the transcriptomic expression levels of BMP target genes between *ACVR1* WT (red) and mutant pDMG (blue) in cohort 1. H3 mutational status is specified as wild-type (WT) (green), mutated on variant H3.1 (brown) or H3.3 (purple). Normalized and centered gene

*Figure 1 continued on next page*

*Figure 1 continued*

expression levels are color-coded with a blue (low expression) to red (high expression) gradient. Samples in columns are clustered using Euclidean distance. The BMP targets gene list is presented in **Supplementary file 1a**. (**C**) TGF-β/BMP pathway activity inferred from a specific genes-response signature using PROGENy algorithm in *ACVR1* WT (red) versus mutant pDMG (blue) (cohort 1). No significant difference was observed between both groups. (**D**) Comparison of the expression level of BMP ligands, antagonists, receptors, and effectors between *ACVR1* WT (red) versus mutant pDMG (blue) in cohort 1. p-values are indicated for each gene. (**E**) Boxplot of BMP ligands expression (vst-normalized) in cohort 1. (**F–G**) Pattern of expression of BMP2 and 7 in developing brain. **Left panel:** heatmaps showing relative BMP2 (**F**) and BMP7 (**G**) expression by in situ hybridization (ISH) in murine brain across development obtained from the ALLEN Developing Mouse Brain Atlas. Normalized and scaled gene expression levels are color-coded with a yellow (low expression) to red (high expression) gradient. Developmental stage is mentioned in rows with pre- and post-natal stages color-coded in dark and light blue, respectively. Different brain regions are indicated in columns as follows: RSP: rostral secondary prosencephalon; Tel: telencephalic vesicle; PedHy: peduncular caudal hypothalamus; P3: prosomere 3; P2: prosomere 2; P1: prosomere 1; M: midbrain; PPH: prepontine hindbrain; PH: pontine hindbrain; PMH: pontomedullary hindbrain; MH: medullary hindbrain (medulla). **Right panel:** spatiotemporal gene expression data of BMP2 (**F**) and BMP7 (**G**) expression from human developing and adult brain samples obtained from the Human Brain Transcriptome (**Hino et al., 2015**). The vertical line indicates birth at 266 days. Each curve represents a part of the brain as following: NCX: neocortex (dark blue); HIP: hippocampus (light blue); AMY: amygdala (orange); STR: striatum (black); MD: mediodorsal nucleus of the thalamus (green, in bold); CBC: cerebellar cortex (red).

The online version of this article includes the following figure supplement(s) for figure 1:

**Figure supplement 1.** Bone morphogenetic protein (BMP) pathway activation coincides with high BMP2/7 levels in *ACVR1* WT-H3.3K27M pDMG.

and *Figure 1—figure supplement 1E, F*). Once induced, BMP ligand production can be maintained by a positive transcriptional regulatory loop (*Kozmikova et al., 2013*; *Christiaen et al., 2010*). We then reasoned that in *ACVR1* WT pDMG, the priming signal may come from the microenvironment. Accordingly, it has been recently suggested that H3.3K27M DIPG likely occur in cells derived from dorsal PAX3+ BMP-reliant progenitors, and that the oncogenic transformation may result from a cross-talk with BMP ligands present in the microenvironment at that time (*Jessa et al., 2022*). Regulation of dorsal glial cell fate during development has been shown to rely mostly on BMP4 and BMP7 (*Liem et al., 1995*; *Wilson et al., 2000*; *Hawley et al., 1995*; *Wilson and Hemmati-Brivanlou, 1995*). By analyzing data from the ALLEN Developing Mouse Brain Atlas (*Lein et al., 2007*), we observed that BMP7 is only expressed at E13.5 during embryogenesis. It is then progressively re-expressed post-natally from P14 notably in the pontine hindbrain, to reach a maximum in most territories including prosomere 2 at P28, which is compatible with the spatio-temporal window of occurrence of H3.3K27M pDMG tumors (*Jessa et al., 2022*; *Figure 1G*). BMP2 is only expressed at the E18.5 embryonic stage and BMP4 expression peaks at P14 corresponding to infancy (*Burford, 2021*), before decreasing during the post-natal period in these territories (*Figure 1F*; *Figure 1—figure supplement 1G*). Similarly, by integrating transcriptomic data from the HBT program (*Kang et al., 2011*), we observed that BMP7 expression but not BMP2 and BMP4 gradually increases to reach its maximum in the midline structure in the mid-childhood period (green curve, *Figure 1F, G* and *Figure 1—figure supplement 1G*), then coinciding with the peak incidence of pDMG tumors (*Mackay et al., 2017*).

Overall, the integration of transcriptomic data reveals that the induction of BMP signaling in *ACVR1* WT pDMG, at a level equivalent to that observed in mutant tumors, could notably result from the production of BMP2 and/or 7, initiated by the expression of BMP7 by the microenvironment.

## BMP7 synergizes with K27M to induce a transcriptomic program leading to quiescence and invasiveness in a low-grade glioma model

To functionally dissect the impact of H3.3K27M/BMP crosstalk and define whether it may have a rather oncogenic or tumor suppressive value, we first used the two previously described pediatric glioma Res259 and SF188 cell lines, which have been genetically modified to stably express and reproduce the epigenetic context either of WT or mutated forms of the variant H3.3 (*Rakotomalala et al., 2021*). Interestingly, BMP7 expression significantly decreases after H3.3K27M induction in both SF188 and Res259 cells (*Figure 2A*), whereas BMP2 and BMP4 expressions are respectively increased or unmodified (*Figure 2A* and *Figure 2—figure supplement 1A, B*). To compensate for that decrease and mimic the high level of BMP7 expression observed in pDMG tumors, we then assessed the impact of recombinant BMP7 depending on the H3.3 context.

Using qRT-PCR as a first hint, we observed a significant increase in BMP targets expression following BMP7 treatment in both H3.3K27M-Res259 and -SF188 cells compared to H3.3 WT cell lines, with at least a twofold increase at 3 and 24 hr for both *ID1* and *ID2* (*Figure 2—figure supplement 1C*).

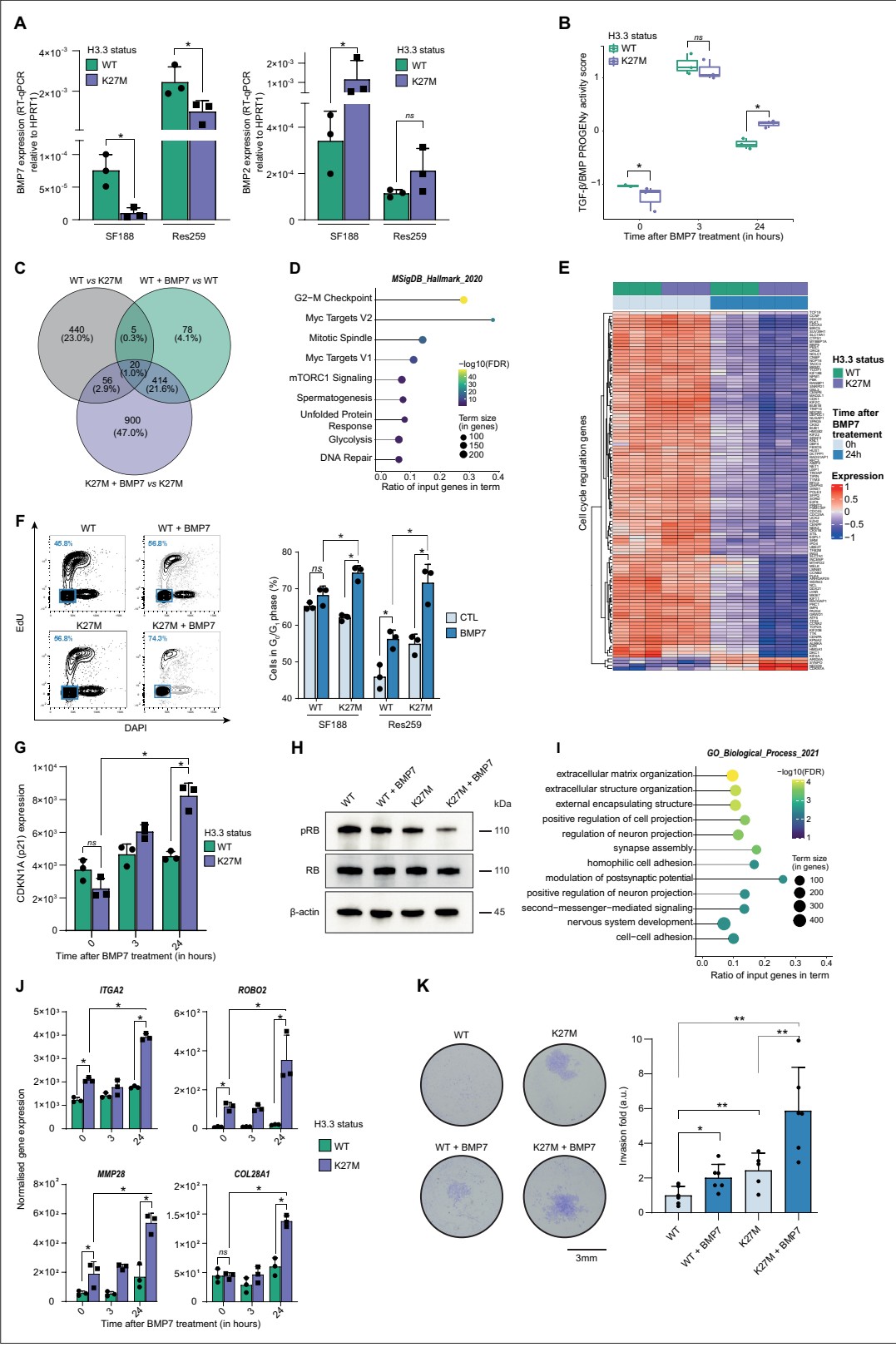

**Figure 2.** BMP7 induces a specific transcriptomic and phenotypic switch in a H3.3K27M mutant glioma context. (**A**) *BMP7* (left) and *BMP2* (right) expressions in H3.3WT (green) versus H3.3K27M (purple) SF188 and Res259 cells. Gene expressions were analyzed by qRT-PCR relative to *HPRT1* expression. Means ± std are represented (n=3). * p<0.05, ns: non-significant. (**B**) TGF-β/BMP pathway activity inferred from a specific genes-response

*Figure 2 continued on next page*

*Figure 2 continued*

signature using the PROGENy algorithm in Res259-H3.3WT (green) and H3.3K27M (purple) after 0, 3 and 24 hr of BMP7 treatment (n=3). *p<0.05, ns: non-significant. (**C**) Venn diagram showing the number of differentially expressed genes (DEG) and the corresponding percentages compared to all DEG in each condition: Res259-H3.3WT versus Res259-H3.3K27M without BMP7 treatment (grey), Res259-H3.3WT versus Res259-H3.3WT treated with BMP7 for 24 hr (green), Res259-H3.3K27M versus Res259-H3.3K27M treated with BMP7 for 24 hr (purple). (**D**) Functional enrichment of DEG specifically between Res259-H3.3K27M versus Res259-H3.3K27M treated with BMP7 for 24 hr. Dots are colored according to their false discovery rate with a blue (lower significance) to yellow (higher significance) gradient and sized by the count number of genes matching the biological process. (**E**) Heatmap representing the transcriptomic expression levels of genes associated with cell cycle regulation between Res259-H3.3WT (green) and Res259-H3.3K27M (purple) cells, with (dark blue) or without (light blue) BMP7 treatment. Normalized and centered gene expression levels are color-coded with a blue (low expression) to red (high expression) gradient. (**F**) Flow cytometry analyses of cell cycle in Res259/SF188-H3.3WT and H3.3K27M upon BMP7 treatment. **Left panel:** representative density plots with outliers (dots) with 5-ethynyl-2'-deoxyuridine (EdU) staining on the y-axis and with DAPI staining on the x-axis for the indicated conditions on Res259 cell lines. Quantification of cells in G0/G1 phase (blue square, low EdU, and low DAPI stainings) appear in the lower left corner for the presented graph. **Right panel:** quantification of cells in G0/G1 phase for SF188- and Res259-H3.3WT or H3.3K27M without BMP7 treatment (light blue) or after 24 hr treatment (dark blue). Means ± std are represented (n=3). *p<0.05, ns: non-significant. (**G**) *CDKN1A* (encoding p21) normalized expression from transcriptomic data of Res259-H3.3WT (green) and Res259-H3.3K27M (purple) after 0, 3, or 24 hr of BMP7 treatment. Means ± std are represented (n=3). *p<0.05, ns: non-significant. (**H**) Western-blot analysis of RB phosphorylation on S780 (pRB) in Res259-H3.3WT or H3.3K27M upon BMP7 treatment. Total RB and β-actin are used as controls. One representative experiment out of 3 is shown. (**I**) Functional enrichment of DEG specific for the K27M/BMP7 condition, according to the decision tree algorithm presented in *Figure 2—figure supplement 1J*. Dots are colored according to their false discovery rate with a blue (lower significance) to yellow (higher significance) gradient and sized by the count number of genes matching the biological process. (**J**) *ITGA2, ROBO2, MMP28,* and *COL28A1* normalized expression from transcriptomic data of Res259-H3.3WT (green) and Res259-H3.3K27M (purple) after 0, 3 or 24 hr of BMP7 treatment. Means ± std are represented (n=3). *p<0.05, ns: non-significant. (**K**) Impact of BMP7 treatment on invasion in Res259-H3.3WT versus H3.3K27M. **Left panel:** representative pictures of a transwell invasion assay of Res259-H3.3WT or H3.3K27M, with and without BMP7 treatment. Scale bar = 3 mm. **Right panel:** invasion was quantified as the mean value of five independent experiments and represented as a graph. Means ± std are represented. *p<0.05, **p<0.01.

The online version of this article includes the following source data and figure supplement(s) for figure 2:

**Source data 1.** Uncropped and labeled gels for *Figure 2*.

**Source data 2.** Raw unedited gels for *Figure 2*.

**Figure supplement 1.** H3.3K27M mutant context potentiates BMP7-induced transcriptomic and phenotypic switch in a glioma model.

**Figure supplement 1—source data 1.** Uncropped and labeled gels for *Figure 2—figure supplement 1*.

**Figure supplement 1—source data 2.** Raw unedited gels for *Figure 2—figure supplement 1*.

Interestingly, both levels and kinetics of SMAD1/5/8 phosphorylation in H3.3K27M versus WT over-expressing cells remain unchanged, indicating that BMP canonical pathway activation is not modified by histone mutational status (*Figure 2—figure supplement 1D*). To further investigate and characterize the specificity of the response of H3.3 WT and mutant cells to BMP7, we performed an RNA-sequencing (RNA-seq) analysis after 3 hr and 24 hr of treatment with recombinant BMP7. Using PROGENy analysis, we observed that the increase in TGF-β/BMP activation score induced by BMP7 is potentiated and remains significantly higher at 24 hr in H3.3K27M-Res259 compared to their wild-type counterparts (*Figure 2B*). While differentially expressed genes at 3 hr mostly correspond to the expression of the K27M mutation (*Figure 2—figure supplement 1E, F*), a subset of 900 genes appears differentially expressed (DE) specifically between treated and untreated H3.3K27M-Res259 cells, but not in H3.3 WT ones (*Figure 2C*). Enrichment analyses revealed that DE genes are notably associated with alteration in cell cycle regulation (*Figure 2D–E*), and that the downregulated ones in H3.3K27M BMP7-treated cells correspond to E2F targets (*Figure 2—figure supplement 1G*). Consistently, BMP7 treatment leads to a significant 12.2% and 16.7% increase of cells in the G0/G1 phase, respectively in H3.3K27M-SF188 and -Res259 cells, while its effect is limited to a 3% and 10.3% increase in their WT counterparts (*Figure 2F*). Similarly, the number of H3.3K27M-Res259 cells is significantly reduced by

1.7-fold compared to non-treated cells upon BMP7 treatment, while the decrease is limited to 1.2-fold in WT ones (*Figure 2—figure supplement 1H*). Of note, this K27M-dependent BMP7 effect is associated with a significant 1.8-fold increase in cyclin dependent kinase inhibitor 1 A (*CDKN1A*, encoding P21; *Figure 2G*) expression and a reciprocal decrease in RB1 phosphorylation (*Figure 2H*). This cell cycle blockade is unlikely to result from the entry of cells in senescence since there is no difference in beta-galactosidase activity (SA-β-gal) between wild-type and mutant cells upon BMP7 treatment (*Figure 2—figure supplement 1I*).

To further dissect the crosstalk between BMP7 and the H3.3K27M mutation, we established a decision tree algorithm to specifically isolate which of the 900 DE genes post-treatment BMP7 in H3.3K27M cells correspond to a potentiation of the effect of the K27M mutation by BMP7 (*Figure 2—figure supplement 1J*, left panel), or to a specific effect of BMP7 in the K27M context (*Figure 2—figure supplement 1J*, right panel). We then pinpointed DE genes that specifically correspond to cooperative effects of K27M epigenetic alterations and BMP7-mediated transcriptional regulation. Interestingly, enrichment analyses revealed that these genes are involved in processes related to invasion/migration, including extracellular matrix organization, regulation of cell/neuron projection, and adhesion (*Figure 2I*). Some of these genes such as *ITGA2*, *ROBO2*, and *MMP28* are already induced following H3.3K27M expression, and their expression is further amplified by the BMP7 treatment (*Figure 2J*). Conversely, others, such as *COL28A1,* are specifically induced or repressed by the K27M+BMP7 context (*Figure 2J*), consistently with our decision tree algorithm. We then assessed the combined impact of H3.3K27M expression and exposure to BMP7 on the invasive properties of glioma cells using a Matrigel-coated transwell assay (*Figure 2K*). Expression of the H3.3K27M mutation is sufficient to drive a moderate increase in invasion compared to the WT context. However, this phenomenon is largely amplified by BMP7, with a global 5.6-fold in the H3.3 mutant context, compared to twofold in the H3.3 WT one.

Altogether, these data support the fact that BMP7 is sufficient to induce a transcriptomic reprogramming specific to the H3.3K27M epigenetic context, which leads to the emergence of a quiescent but invasive cell state.

## Combined BMP2/BMP7 expression drives a quiescent-invasive tumor cell state in pDMG

Considering the data obtained in the Res259/SF188 mechanistic models, we then sought to define whether this crosstalk between BMP and H3.3K27M was preserved with similar effects in H3.3K27M pDMG models. First, we observed that BMP7 is the most expressed ligand in two *ACVR1* WT/H3.3K27M DIPG cell lines, but that few if any other BMP ligands, including BMP4, are (*Figure 3—figure supplement 1A*). Unlike in tumors, BMP2 expression is notably low in these cell lines (*Figure 3—figure supplement 1A*). BMP2 has been shown to be induced by hypoxia (*Tseng et al., 2010*) or reactive oxygen species (ROS) (*Sánchez-de-Diego et al., 2019*). Accordingly, exposure of *ACVR1* WT H3.3K27M DIPG cells to hypoxia or ONC201, which is known to significantly increase ROS production (*Przystal et al., 2022*; *Wu et al., 2023*), are both sufficient to induce a significant increase in BMP2 expression (*Figure 3A*). Thus, BMP2 can be produced autonomously by tumor cells in response to specific stresses. To model the concomitant impact of BMP7 and stress-induced BMP2 in the H3.3K27M mutant background, we analyzed the impact of BMP2 addition on DIPG 3D spheroids. As shown in *Figure 3B* and *Figure 3—figure supplement 1B, C*, treatment of DIPG spheroids with increasing doses of recombinant human BMP2 triggers SMAD1/5/8 phosphorylation and leads to a strong dose-dependent decrease in growth rate. Consistently, Ki67 staining, a marker of proliferation, is significantly decreased upon BMP2 treatment (*Figure 3C*). Reciprocally, treatment of DIPG-spheroids with the BMP inhibitor LDN-193189 (LDN) leads to a slight increase in KI67-positive cells (*Figure 3C*). This effect was largely mitigated in BT245 and DIPGXIII cell lines in which the K27M mutant allele was removed (*Harutyunyan et al., 2019*), indicating that it may depend on the K27M-specific epigenetic context (*Figure 3—figure supplement 1D*). Of particular interest, the combined knock-out of K27M and BMP inhibition with LDN treatment resulted in the failure of pDMG cells to proliferate and form gliospheres (*Figure 3—figure supplement 1D*). In parallel, we explored the impact of BMP activation/inhibition on DIPG cells migration/invasion propensity. BMP2 significantly increases the migration of tumor cells from Matrigel-embedded 3D-DIPG spheroids, while an antagonistic effect was observed upon LDN treatment (*Figure 3D* and *Figure 3—figure supplement 1E*).

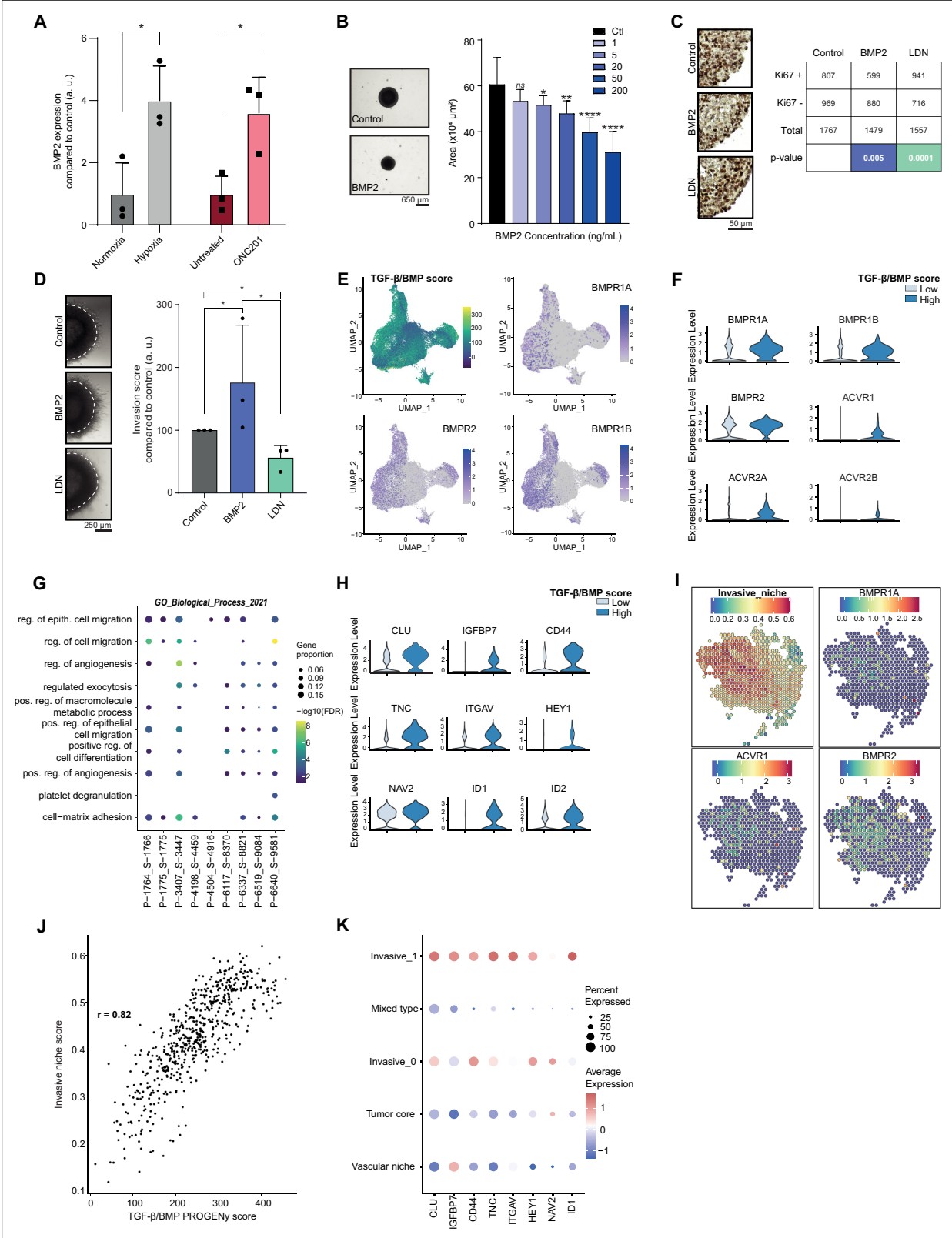

**Figure 3.** Combined tumor-autonomous BMP2/BMP7 expression drives a quiescent-invasive tumor cell state in pediatric diffuse midline gliomas (pDMG). (**A**) *BMP2* expression after hypoxia or ONC201 treatment. Gene expression was analyzed by QRT-PCR relative to *HPRT1* expression. Means ± std are represented (n=3). *p<0.05. (**B**) Growth monitoring of HSJD-DIPG-012 following recombinant BMP2 treatment. Means± std are represented (n=3). *p<0.05, **p<0.01, ****p<0.0001, ns: non-significant. Scale bar = 650μm. (**C**) Impact of BMP2 or LDN treatment on KI67-positive staining in

*Figure 3 continued on next page*

*Figure 3 continued*

HSJD-DIPG-012. **Left panel:** representative images of Ki67 immunohistochemistry on HSJD-DIPG-012 spheroids treated or not with either BMP2 or LDN-193189. Scale bar = 50 μm. **Right panel:** quantification of Ki67-positive and negative cells. p-values were computed using Fisher's exact test. BMP2: 200 ng/mL, LDN-193189: 1 μM. (**D**) Impact of BMP2 or LDN treatment on tumor cells invasion. **Left panel:** representative images of HSJD-DIPG-012 spheroids embedded in Matrigel, after 48 hr of BMP2 or LDN-193189 treatment. Scale bar = 250 μm. **Right panel:** Invasion was quantified as the mean value of four independent experiments and represented as a graph. *p<0.05. BMP2: 10 ng/mL, LDN-193189: 1 μM. (**E**) UMAP (uniform manifold approximation and projection) computed on harmony embeddings of the tumor cells of 10 *ACVR1* WT-H3.3K27M pDMGs from the scRNA-seq data published by *Jessa et al., 2022*. PROGENy TGF-β/BMP score is colored from blue (low activity score) to yellow (high activity score). Expression of the bone morphogenetic protein (BMP) receptors *BMPR1A, BMPR2* and *BMPR1B* is colored from grey (low expression values) to purple (high expression values). (**F**) Violin plots of BMP receptors in one (P-1775_S-1775) of the 10 samples based on the TGF-β/BMP PROGENy score. The TGF-β/BMP-low and high groups are colored respectively in light and dark blue. All p-value are lower than 7.357206e-13 (*ACVR2A*). (**G**) Dotplot of the first 10 significantly enriched pathways (FDR ≤ 0.05) in TGF-β/BMP-high cells for each of the 10 *ACVR1* WT-H3.3K27M pDMGs, ranked by number of samples with a significant enrichment, using the GO Biological Process database. Only dots of significant enrichments are shown for each sample. Dot color represents the -log10 (p-value) and ranges from blue (high p-value) to yellow (low p-values). Dot size is proportional to the overlap of differentially expressed (DE) genes and the genes of a geneset. (**H**) Violin plots of invasion-related genes in one (P-3407_S-3447) of the 10 samples based on the TGF-β/BMP-high/low score. The TGF-β/BMP-low group and high group are respectively colored in light and dark blue. All p-value are lower than 1.34e-8 (*HEY1*). (**I**) 'Invasive niche' score from *Ren et al., 2023*, and associated expression of the BMP receptors *BMPR1A, ACVR1,* and *BMPR2*. Color ranges from blue (low score/expression value) to red (high score/expression value). (**J**) Scatter plot correlating the PROGENy TGF-β/BMP pathway activity score with the 'Invasive niche' score for each Visium spot of pDMG Sample-1. The correlation coefficient was computed using Pearson's method. ****p-value < 2.e-16. (**K**) Scaled expression of invasion-related genes in pDMG Sample-1 for each identified area. Dot size represents the proportion of cells expressing the gene. Color ranges from blue (low expression) to red (high expression).

The online version of this article includes the following source data and figure supplement(s) for figure 3:

**Figure supplement 1.** Combined tumor-autonomous BMP2/BMP7 expression drives a quiescent-invasive tumor cell state in pediatric diffuse midline gliomas (pDMG).

**Figure supplement 1—source data 1.** Uncropped and labeled gels for *Figure 3—figure supplement 1*.

**Figure supplement 1—source data 2.** Raw unedited gels for *Figure 3—figure supplement 1*.

To define whether such a BMP-induced quiescent-invasive cell state exists in *ACVR1* WT H3.3K27M pDMG tumors, we first used publicly available scRNA-seq data from 10 patients' biopsies from *Jessa et al., 2022*. Integration of all these data unveiled that the BMP-responsive (i.e. High PROGENy TGF-β/BMP score) pool of tumor cells correlates significantly with BMP receptors expression, in particular to *BMPR2, BMPR1B, BMPR1A,* and *ACVR1* levels, respectively in 8, 5, 5, and 4 out of 10 samples (*Figure 3E–F* and *Figure 3—figure supplement 1F*). Interestingly, this BMP-responsive pool of cells is significantly enriched in genes involved in positive regulation of cell migration and in cell-matrix adhesion (*Figure 3G*), while showing a specific quiescent-compatible decrease in genes involved in transcription/translation processes (*Figure 3—figure supplement 1G*). To further define the extent and organization of this pool of cells, we performed spatial transcriptomic analysis of 3 H3.3K27M pDMG tumors. Data integration was performed using pDMG-derived Visium and scRNA-seq signatures recently published by *Ren et al., 2023* and *Filbin et al., 2018* (*Figure 3—figure supplement 1H*). Consistently with observations in bulk and single-cell analyses of patients' tumors/models, the most highly expressed BMP ligands are BMP2 and BMP7 along with BMP8B (*Supplementary file 1b* and *Figure 3—figure supplement 1H*), whose expression is not spatially delimited or associated with a specific gene signature. In line with single-cell analyses, we observed on the contrary that BMP receptors expression correlate significantly with the invasive niche score defined by *Ren et al., 2023* (*Figure 3I*), which can be spatially restricted to histologically delineated areas (samples 1–3, *Figure 3I* and *Figure 3—figure supplement 1H*) or more dispersed within the tumor (sample 2, *Figure 3—figure supplement 1H*). Moreover, the invasive niche score strongly correlates to the PROGENy TGF-β/BMP score (*Figure 3J* and *Figure 3—figure supplement 1I*). Coherently, this BMP-responsive invasive niche is characterized by a high expression of key markers of BMP activity (*ID1*), as well as markers of stemness (*CD44, HEY1*) and migration/invasion (*TNC, IGFBP7, ITGAV*) (*Figure 3K*).

Altogether, these data indicate that tumor-autonomous production of BMP2 and BMP7 synergize to maintain a quiescent-invasive niche in H3.3K27M DIPG.

## Discussion

Major advances have been made in understanding the molecular bases of pDMG, among which DIPG, with the identification of major epigenetic remodeling processes induced by histone H3 mutations. However, the oncogenic mechanisms cooperating with these mutations to induce transformation and tumor escape remain largely undefined.

Along this line, a key challenge is to establish the precise role of the transcriptomic reprogramming induced by the BMP pathway, which remains controversial in these pediatric brain tumors (*Carvalho et al., 2019*; *Fortin et al., 2020*; *Hoeman et al., 2019*; *Jessa et al., 2022*; *Sun et al., 2022*). Herein, we performed a comprehensive integration of transcriptomic data, which first supports the view of BMP signaling being also clearly activated in *ACVR1* WT/H3.3K27M pDMG. Further analyzes will be necessary to define whether the lowest levels of activation correspond to samples with specific locations along the midline. Effect of BMP activation is potentiated by the epigenetic context of these tumors, then leading to a global transcriptional reprogramming. Although it has been previously described that *ACVR1*-mutant tumors exhibit a higher expression of *ID1* and *ID2* (*Carvalho et al., 2019*), the extrapolation of BMP activity from a larger signature of BMP targets and on a score calculated from the inference of gene expression perturbations in response to the TGF-β/BMP pathway indicates the existence of a compensatory mechanism driving BMP activation in non-*ACVR1* mutant tumors. By analyzing the expression of BMP effectors in pDMG tumors and patient-derived DIPG models, we propose that the activation of this pathway in *ACVR1* WT H3.3K27M tumors could be at least partially mediated by two complementary tumor-autonomous and microenvironment-dependent mechanisms. Upon initiation, the expression of BMP, and notably BMP7, could synergize with or even trigger the autocrine production of BMP ligands by the tumors, among which BMP2 and 7. Indeed, such positive feedback loops maintaining the expression of BMP ligands have already been described notably during development (*Kozmikova et al., 2013*; *Christiaen et al., 2010*). The fact that the expression of BMP ligands is similar in *ACVR1* mutant and WT tumors and independent of H3 status (*Figure 1—figure supplement 1F*) suggests that this regulatory loop is involved in most tumors, but probably by different mechanisms depending on the mutational context. Thus, BMP secretion by the microenvironment may prime the BMP activation in tumor cells and be required for oncogenic transformation. Once established, the dynamic modulation of BMP2 expression in response to stresses, such as hypoxia or treatments, could synergize with constitutive production of BMP7 to drive the emergence of an aggressive cell state. Of note, the expression of BMP-target genes *ID1* to *4* was previously reported to be strongly decreased in pDMG cell lines in which the K27M mutation was removed by CRISPR-Cas9 (*Harutyunyan et al., 2019*), suggesting that even the maintenance of BMP activation in a pool of tumor cells relies on the specific K27M-mediated epigenetic context. In addition, blocking the BMP pathway with LDN in these K27M-KO cells induces tumor cell death, supporting the view that the K27M/BMP oncogenic synergy plays a major role in maintaining oncogenic potential (*Figure 3—figure supplement 1*).

Second, our data are in favor of a rather global oncogenic role of the BMP pathway in pDMG gliomagenesis. The tumor suppressor activity of the BMP pathway had been largely extrapolated because of its positive impact on tumor cell quiescence (*Sun et al., 2022*). Indeed, using a genetically engineered glioma model, we confirmed that BMP7 is sufficient to potentiate the entry of cells in a quiescent state in a H3.3K27M-dependent manner, via a transcriptomic switch largely relying on the downregulation of E2F-targets cell-cycle regulating genes. Nevertheless, this quiescent cellular phenotype could paradoxically constitute an aggressive treatment-resistant state, as previously observed in adult glioblastoma (*Atkins et al., 2019*; *Chen et al., 2012*; *Xie et al., 2022*) and thus explaining its increase in response to treatment (*Sun et al., 2022*). Along the same line, the impact of *CHRDL1* increase in H3.3K27M tumors on BMP pathway inhibition may probably need to be qualified (*Sun et al., 2022*). Indeed, if CHRDL1 was first classified as a member of the chordin family of secreted BMP antagonists due to sequence homology, it has been shown that it exerts BMP-independent functions in synapse plasticity and maturation (*Blanco-Suarez et al., 2018*). Moreover, this protein has also been described as an activator of the BMP pathway (*Lin et al., 2005*), suggesting that its high level of expression may not be a robust marker of BMP activation state.

However, it cannot be ruled out also that different ligands of the BMP pathway may have different impacts on the cellular phenotype induced by the H3.3K27M mutation. Accordingly, it was shown that BMP4 treatment promotes the differentiation of DIPG tumor cells, in line with its putative tumor

suppressor activity (*Sun et al., 2022*). However, beyond quiescence, we observed that BMP activation by BMP2/7 in a H3.3K27M epigenetic context induces a transcriptomic switch rather than conferring enhanced invasion potential to pDMG tumor cells. Because the level of BMP2/7 is particularly important (i) in bulk, single-cell and spatial transcriptomic tumors, (ii) in patient-derived cell models, and because the dynamics of BMP7 induction in the post-natal period coincide with the spatio-temporal window of pDMG onset, we believe that the role of the BMP2/7 couple is non-negligible in the pathogenesis of pDMG. Interestingly, this pair has already been shown to trigger BMP signaling as a heterodimer notably during embryogenesis, with a higher efficiency than homodimers (*Kim et al., 2019*; *Schmid, 2000*; *Tajer et al., 2021*). Keeping in mind that BMP2 appears to be dynamically regulated by tumor cells upon stress, these data suggest that activation of the BMP pathway may be finely regulated in pDMG to be maintained at optimal pro-oncogenic levels, without triggering the tumor-suppressive effects (*Sun et al., 2022*) or negative regulatory loops associated with its over-activation (*Christiaen et al., 2010*; *Afrakhte et al., 1998*; *Akizu et al., 2010*; *Bénazet et al., 2009*).

The next question is whether a therapeutic perspective can be defined by targeting the crosstalk between epigenetic modifications induced by the H3.3K27M mutation and transcriptomic reprogramming induced by BMP2/7. The BMP2/7-driven cell state that we described here fits with previous results obtained in other glioma models, in which a subpopulation of quiescent cells was identified as partially responsible for tumor invasiveness (*Atkins et al., 2019*; *Antonica et al., 2022*; *Furst et al., 2022*), a hallmark of pDMG aggressiveness (*Kluiver et al., 2020*). Given the complexity of the phenotype induced by H3.3K27M/BMP crosstalk and the pleiotropic role of BMP proteins in the central nervous system, the therapeutic strategy to be developed should be based on the targeting of the downstream effectors, such as ID1 as recently described (*Messinger et al., 2023*), or TNC, which may be responsible for the invasive phenotype. Upcoming challenges will be to precisely define the identity of these pro-invasive BMP effectors, to set up a combinatorial therapeutic approach, simultaneously targeting the proliferative compartment and the BMP-responsive H3.3K27M invasive cell state.

# Materials and methods
## Gene expression analyzes of publicly available transcriptomic datasets
### Gene-expression analyzes of H3.3K27M-pDMG transcriptomic cohorts
For cohort 1 (*McLeod et al., 2021*), HTSeq gene counts and somatic vcf of St Jude's pDMG samples were downloaded from https://platform.stjude.cloud/. DESeq2 (v 1.36) (*Love et al., 2014*) was used to normalise the data with the variant stabilization transformation (vst) (*Anders and Huber, 2010*). *ACVR1* mutation status was assessed using tabix (*Li, 2011*) (v 1.15.1) to query the region of the *ACVR1* gene chr2:157,736,251–157,876,330 (hg38). Identified variants were manually curated. PROGENy (v 1.18.0) (*Schubert et al., 2018*) was used to infer TGF-β/BMP pathway activity with the 'top' parameter set to 100 (default value).

For cohort 2 (*Mackay et al., 2017*), gene expression (z-score) and tumor variants were downloaded on https://pedcbioportal.kidsfirstdrc.org/ from the dataset 'phgg_jones_meta_2017.' Samples were filtered based on the following location: brainstem or midline, and the following histone mutation: WT, H3.1K27M, and H3.3K27M. Only the four datasets with *ACVR1* mutant samples were kept for the analysis to avoid biases: 'PMID:21931021|PMID:24705251,' 'PMID:21931021|PMID:22286216,' 'PMID:22389665|PMID:24705252,' 'PMID:24705251.' TGF-β/BMP pathway activity was also inferred using PROGENy, but the 'top' parameter was set to 178 genes: as half of the top 100 genes of the PROGENy model for the TGF-β/BMP pathway are not covered in cohort 2, we identified the top 100 genes with the highest absolute coefficient in PROGENy's model to compute pathway activity with the same number of genes in both cohorts.

For both cohorts, PCA was performed using FactoMineR (v.2.4) (*Lê et al., 2008*) and plotted with factoextra (v1.0.7) (*Kassambara and Mundt, 2017*).

### Patterning of BMP expression in the developing brain
Spatio-temporal gene expression data of *BMP2*, *BMP4* and *BMP7* from human developing and adult brain samples were obtained from the Human Brain Transcriptome project (https://www.hbatlas.org) (*Kang et al., 2011*). Heatmaps showing relative *BMP2*, *BMP4,* and *BMP7* expression by in situ

hybridization (ISH) in murine brain across development were obtained from the ALLEN Developing Mouse Brain Atlas (https://www.developingmouse.brain-map.org; *Lein et al., 2007*).

### scRNA-seq analyses of H3.3K27M ACVR1 WT DMG

Using the scRNA-seq data published by *Jessa et al., 2022* (GSE210568) , we selected H3.3K27M-*ACVR1* WT pDMGs localized either in the pons or in the thalamus (10 samples). The same processing steps as the authors were performed except for the regression of mitochondrial proportion and the number of UMIs. Normalization was performed with the LogNormalize function of Seurat (*Hao et al., 2021*). All samples were filtered to keep tumoral cells as identified by the authors. As a readout of BMP pathway activity, we used PROGENy to infer the TGF-β/BMP score. We then stratified cells into TGF-β-High and TGF-β-Low groups using the 95$^{th}$ quantile as the cutoff. FindMarkers with default parameters was used to identify differentially expressed (DE) genes (FDR ≤0.05) in each group. Enrichment analyzes were then conducted using the enrichR package (*Kuleshov et al., 2016*) and the Gene Ontology Biological Process gene signatures (*Ashburner et al., 2000*), separately on DE genes upregulated in TGF-β-High and TGF-β-Low for each sample. Only the top 10 significant enrichments (FDR ≤0.05) are shown on the plots. Identification of DE BMP ligands and receptors was run with Find-Markers using a logfc.threshold of 0.15 (instead of the default 0.25). For the visualization of all samples on a shared UMAP, Harmony (*Korsunsky et al., 2019*) (v 0.1.1) was used to integrate the 10 samples on the 50 principal components computed. The first 30 harmony-corrected principal components were then used to compute the SNN graph and UMAP.

## RNA-sequencing of in-house pDMG samples

As part of the Share4Kids program, a third cohort was constituted from leftover DMG samples, obtained through biopsies performed at the Pediatric Hematology and Oncology Institute (iHOPE, Lyon) and the Hôpital Femme Mère Enfant (HFME, Lyon). Tissue banking and research were conducted according to national ethics guidelines, after obtaining the written informed consent of patients. This study was approved by the ethical review board of the BRC of the Centre Léon Bérard (noBB-0033–00050, No 2020–02). This BRC quality is certified according to AFNOR NFS96900 (No 2009/35884.2) and ISO 9001 (Certification N° 2013/56348.2). Biological material collection and retention activity are declared to the Ministry of Research (DC-2008–99 and AC-2019–3426). For RNA-seq library construction of cohort 3, total RNAs from tissues were isolated using the AllPrep DNA/RNA FFPE kit (Qiagen, 80224) following the manufacturer's instructions. Libraries were prepared with Illumina Stranded mRNA Prep (Illumina, 20040534) following recommendations. Quality was further assessed using the TapeStation 4200 automated electrophoresis system (Agilent) with High Sensitivity D1000 ScreenTape (Agilent). All libraries were sequenced (2×100 bp) using Agilent SureSelect RNA XTHS2 All Exon V8 (Agilent, G9991A) according to the standard Agilent protocol.

Quality control of reads was performed using FastQC (v.0.11.9) (*Andrews, 2010*), followed by trimming of Illumina adapter sequences with Cutadapt (*Martin, 2011*) (v.3.4) using the -a CTGTCTCT TATACACATCT and -A CTGTCTCTTATACACATCT parameters. Reads were mapped to the GRCh38 human genome using 'two-pass' mode STAR (v.2.7.9) (*Dobin et al., 2013*), with Ensembl v104 annotations. Gene counts were then computed using HTseq-count (v.0.13.5) (*Anders et al., 2014*) with the following parameters: '--order pos' and '--stranded reverse.' HTseq count files were then loaded in R (v 4.2.0) and two filtration steps were applied using the annotations of org.Hs.eg.db (v 3.15.0) *Carlson et al., 2019*: genes with low counts (less than 10 reads across samples) were removed; non-protein-coding genes were removed. Filtered gene counts were converted into a DESeq2 object with the design parameter set to account for the tumor histone mutation (HGG/DIPG) and normalized with vst using 'blind = FALSE.'. The R package ggplot2 (v 3.3.5) (*Villanueva and Chen, 2019*) was used to plot the expression of BMP ligands.

## Spatial transcriptomics analyzes

FFPE tissue sections were placed on Visium slides and prepared according to 10X Genomics protocols. After H&E staining, imaging and decrosslinking steps, tissue sections were incubated with human-specific probes targeting 17,943 genes (10X genomics, Visium Human Transcriptome Probe Set v.1.0). Probes hybridised on mRNA were captured on Visium slides and a gene expression library

prepared following 10X genomics dedicated protocol and sequenced on Illumina NovaSeq 6000 targeting 50,000 reads per spot.

Raw reads were pre-processed using the space ranger count pipeline (v 2.0.0) and the human GRCh38 reference provided by 10X Genomics. Filtered H5 matrices were loaded using the Load10X_ Spatial function from Seurat. The following pipeline was applied to process raw counts: counts were normalized with the LogNormalise method and a scale factor of 10000; the top 4000 variable features ('nfeatures' parameter) were identified using the vst method; data were then scaled for all genes, the first 50 components of the PCA were computed using the 4000 variable features identified; SNN graph was constructed using the first 30 dimensions of the PCA (dims parameter); clusters were identified with the Louvain algorithm along several resolutions ranging from 0.1 to 1; UMAP was also computed using the first 30 dimensions of the PCA (dims parameter). Signatures identified by *Ren et al., 2023* and *Filbin et al., 2018* were used in conjunction with sc-type (*Ianevski et al., 2022*) to annotate the clusters at a resolution of 0.5. Annotated clusters were validated with an anatomopathologist, leading to the sub clusterization of the 'Invasive niche' of HFME-1 into 2 new clusters ('Invasive_niche_1' and 'Invasive_niche_2'). Scoring of Ren's signatures were made with the AddModuleScore function of Seurat. PROGENy was used to score TGF-β/BMP pathway activity. Markers of the 'Invasive niche' were identified with the FindMarkers function using a logfc.threshold=0.15.

## Cell culture and treatments

Human pediatric low-grade glioma cell line Res259 (grade II, diffuse astrocytoma) and high-grade glioma cell line SF188 (grade IV, glioblastoma) were kindly provided by Dr Samuel Meignan. Both isogenic cell lines overexpressing H3.3 wild-type and H3.3K27M were generated as previously described by *Rakotomalala et al., 2021*. All cell lines were grown as a monolayer in DMEM medium with GlutaMAX-I, 4,5 g/L D-Glucose and 110 mg/L pyruvate (Gibco, 31966) supplemented with 10% foetal bovine serum (FBS), 1 X MEM non-essential amino acid solution (Gibco, 11140050) and 1 X Penicillin-Streptomycin (Gibco, 15140122). Cells were incubated in a humid atmosphere at 37 °C with 5% CO2.

To assess the impact of BMP7 treatment on wild-type or H3.3K27M-mutated Res259 or SF188 cell lines, $1.5x10^5$ cells were plated into six-well plates in a complete medium. After 24 hr, cells were rinsed with PBS and fresh medium and serum-deprived in 1% FBS medium for 3 hr. 50 ng/mL of human recombinant BMP7 (Peprotech, 120–03 P) was then directly added to the medium, and cells were harvested at indicated time points for further experiments.

For pDMG cell lines grown as spheroids, HSJD-DIPG-007, HSJD-DIPG-012, HSJD-DIPG-013, and HSJD-DIPG-014 were kindly provided by Dr Angel Montero-Carcaboso. BT245 and SU-DIP-GXIII, KO or not for H3.3K27M mutation, were kindly provided by Dr Nada Jabado (*Harutyunyan et al., 2019*). pDMG cell lines were grown in a complete culture medium as described by *Monje et al., 2011*, in 96-well ULA plates (Corning, 7007) or 25 cm² low attachment culture flasks (Corning, 431463). The Medium was changed twice a week and spheroids were splitted every 1–2 weeks when reaching 800–1000 µm of diameter using TrypLE Express Enzyme (Thermo Fisher Scientific, 12605010) preheated to 37 °C. Hypoxia and ONC201 treatments were induced on HSJD-DIPG-012 spheroids once a diameter of 600 µm was reached. To induce hypoxia, HSJD-DIPG-012 spheres were incubated for 3 hr in a dedicated incubator at 37 °C, 1% $O_2$, while the controls were incubated at 37 °C in normoxia conditions. Spheroids were similarly treated or not with 20 µM of ONC201 (Selleckchem, S7963) for 96 hr. Exploration of BMP2 effect on HSJD-DIPG-012 and HSJD-DIPG-014 spheroids was done using 10 ng/mL of human recombinant BMP2 (Peprotech, 120–02 C) before protein quantification by western blot.

All cultures were tested every month for mycoplasma using the MycoAlert Mycoplasma Detection Kit (Lonza, LT07-318), in accordance with the manufacturer's instructions.

## RNA-seq data processing and analyses of Res259 cells

For RNA-seq library construction, 1000 ng of total RNAs were isolated using the Nucleospin RNA kit (Macherey-Nagel, 740955) following the manufacturer's instructions. Libraries were prepared with Illumina Stranded mRNA Prep (Illumina, 20040534) following recommendations. Quality was further assessed using the TapeStation 4200 automated electrophoresis system (Agilent) with High Sensitivity

D1000 ScreenTape (Agilent). All libraries were sequenced (2×75 bp) using NovaSeq 6000 (Illumina) according to the standard Illumina protocol.

Raw sequence quality was assessed using FastQC (0.11.9) (*Andrews, 2010*). The trimming step was omitted, as 5' and 3' read bases had a quality greater than Q30, and no adapter fragments were detected. Reads were then pseudo-aligned using Kallisto (0.46.2) (*Bray et al., 2016*) with Ensembl v96, human genome build GRCh38. The rest of the analyzes were performed using R version (4.2.0). Differential expression (DE) analyzes were conducted using the DESeq2 package (1.36.0) (*Love et al., 2014*) using default parameters. Genes with corrected *p*-value (Benjamini–Hochberg)<0.05 and |log2FoldChange (LFC)|>log2(1.5) were considered DE. EnrichR package (3.0) (*Kuleshov et al., 2016*) was used for overrepresentation analysis using the following databases: Gene Ontology (GO) Biological Process 2021, GO Molecular Function 2021, GO Cellular Compartment 2021, KEGG 2021 Human, MSigDB Hallmarks 2020 and ChEA 2016. Enrichment significance was assessed by Fisher's exact test, and *p*-values were corrected with the Bonjamini-Hochberg method. Overrepresentation analyses were run separately on upregulated and downregulated DE genes. PCA was performed using Facto-MineR and plotted with factoextra. Boxplots and heatmaps were respectively made with ggplot2 and ComplexHeatmap (v 2.12.1) (*Gu et al., 2016*) using the vst normalized expression. TGF-β pathway activity was inferred with PROGENy using the first 100 genes of the model (top = 100).

## qRT-PCR profiling

Total RNA was extracted using a Nucleospin RNA kit (Macherey-Nagel, 740955) or RNeasy Plus Micro (Qiagen, 74034) following the manufacturer's instructions. 500–1000 ng was reverse transcribed using the iScript cDNA Synthesis kit (Bio-Rad, 1708891) according to the manufacturer's instructions. Expression of *BMP2* [forward primer (5'-TGCGGTCTCCTAAAGGTCG-3') and reverse primer (5'-GAATTCAGAAGCCTGCAAGG-3')], *BMP7* [forward primer (5'-GGGTGGGTCTCTGTTTCAG-3') and reverse primer (5'-CCTGGAGCACCTGATAAACG-3')], *ID1* [forward primer (5'-GGTGCGCTGTCTGTCTGAG-3') and reverse primer (5'-TGTCGTAGAGCAGCACGTTT-3')] and *ID2* [forward primer (5'-CCCAGAACAAGAAGGTGAGC-3') and reverse primer (5'-GAATTCAGAAGCCTGCAAGG-3')], were assessed by real-time quantitative QRT-PCR on a LightCycler 480 instrument (Roche) using the LightCycler 480 SYBR Green I Master Mix (Roche, 04707516001), according to the manufacturer's instructions. *HPRT1* [forward primer (5'-AAGAGCTATTGTAATGACCAGT-3') and reverse primer (5'-CAAAGTCTGCATTGTTTTGC-3')] expression was used as a housekeeping gene.

The expression of a panel of 84 genes of the TGF-β/BMP signaling pathway was assessed in HSJD-DIPG-007, HSJD-DIPG-012, and HSJD-DIPG-014 cell lines by real-time quantitative QRT-PCR on a LightCycler 480 instrument (Roche) using an RT² Profiler PCR Array (Qiagen, 330231), according to the manufacturer's instructions. RNA were extracted using the RNeasy mini kit (Qiagen, 74104), and 500 ng of RNA were reverse transcribed using the RT2 First Strand Kit (Qiagen, 330401) according to the manufacturer's instructions.

## Western blot

Cells were lysed in RIPA Buffer (50 mM Tris-HCl pH 8, 150 mM NaCl, IGEPAL 1%, 0.5% sodium deoxycholate, 0.1% SDS) containing a protease and phosphatase inhibitor cocktail (Thermo Fisher Scientific, 78440). Protein contents were estimated using the Thermo Scientific Pierce BCA Protein Assay Kit (Fischer Scientific, 23225). Samples were diluted with distilled water to achieve equal concentrations and a loading buffer (4 x Laemmli Sample Buffer, Biorad, 1610747) containing 100 mM DTT (Sigma-Aldrich, 11583786001) was added. Protein extracts were then analyzed by immunoblot. Briefly, proteins were loaded into SDS-polyacrylamide gels for electrophoresis (Mini Protean TGX gels, Biorad, 4561034) and blotted onto polyvinylidene fluoride sheets (Trans-Blot Turbo Transfer PVDF pack, Biorad, 1704156) using the TransBlot technology (Bio-Rad). Membranes were blocked with 5% BSA in TBS-Tween 0.1% for 1 hr and then incubated overnight at 4 °C with anti-pSMAD1/5/8 (1:1000, Cell Signaling Technology, 13820), anti-SMAD1 (1:1000, Cell Signaling Technology, 6944), anti-SMAD1/5/8 (1:1000, Abcam, ab80255), anti-pRb (1:500, Abcam, ab47763), and anti-Rb (1:1000, Cell Signaling Technology, 9309). After three washes with TBS-Tween 0.1%, membranes were incubated with the appropriate HRP-conjugated secondary antibody (1:20.000, Jackson ImmunoResearch). HRP-conjugated β-Actin antibody (1:25.000, Sigma-Aldrich A3854) and HRP-conjugated GAPDH antibody (1:2000 Cell Signaling Technology, 8884), used as loading controls, were incubated for 1 h at room

temperature. After three washes with TBS-Tween 0.1%, the detection was performed using the Amersham ECL Prime Detection Reagent (Cytiva, RPN2232). Membranes were imaged on the ChemiDoc Touch Imaging System (Bio-Rad).

## Proliferation and cell cycle characterization assays

For proliferation assay, $5×10^4$ Res259 or $7.5x10^4$ SF188 cells were plated into six-well plates in a complete medium. After 72 hr of BMP7 treatment, total cell number and viability were quantified by image cytometry on a NucleoCounter NC-3000 (Chemometec) according to the procedure provided by the manufacturer, using a co-staining of Acridine Orange and DAPI (Chemometec, 910–3013).

For cell cycle analyses, $3×10^5$ (*Hoffman et al., 2018*) Res259 or SF188 cells were plated into 10 cm-petri dishes in a complete medium. After 24 hr of BMP7 treatment, 10 µM EdU was added for 1.5 hr. Cells were then harvested using Trypsin (Gibco, 25300054) and washed twice with PBS. Click-it reactions were performed using the Click-iT Plus EdU Alexa Fluor 647 Flow Cytometry Kit (Invitrogen, C10634), according to the manufacturer's instructions. Cells were counterstained with DAPI and analyzed on a BD FACSAria Fusion flow cytometer (BD Biosciences). Data analysis was performed with FlowJo v10.7.1 Software (BD Biosciences). Cells were identified on a Side Scatter (SSC) vs Forward Scatter (FSC) dot plot and cell debris and aggregates were excluded from analysis based on FSC signals.

For β-galactosidase activity determination, $5×10^3$ Res259 cells were plated into 12-well plates in a complete medium. After 72 hr of BMP7 treatment, cells were fixed for 5 min in 0.5% glutaraldehyde, rinsed twice with PBS, and incubated for 48 hr at 37 °C in senescence-associated beta-galactosidase (SA-β-Gal) staining solution as previously described (*Debacq-Chainiaux et al., 2009*). SA-β-Gal + cells were counted manually, and a total number of cells were counted using a DAPI-counterstaining and using the Fiji software (*Schindelin et al., 2012*).

## Invasion assays

For 2D pediatric glioma cell lines (Res259 and SF188), 8.0 µm-Boyden chambers (Falcon, 353097) were coated with 100 µL of 10% Geltrex LDEV-Free, hESC-Qualified, Reduced Growth Factor Basement Membrane Matrix (Gibco, A1413301) completed or not with 50 ng/mL human recombinant BMP7. During the 3 days preceding the experiment, Res259 WT or K27M cells were pre-conditioned or not with 50 ng/mL human recombinant BMP7. $2.5×10^4$ cells were plated over Geltrex in 400 µL 1% FBS-DMEM completed or not with 50 ng/mL human recombinant BMP7. Boyden chambers were deposited in 24-well plates filled with 700 µL of 10% FBS-DMEM. After 48 hr, Boyden chambers were washed with PBS, and incubated for 20 min with methanol to fix the cells, and Crystal violet 0,1% (Sigma-Aldrich, V5265) was used to color cells allowing counting. Pictures of the chambers were taken using an EVOS-7000 (Invitrogen) and analyzed using the Fiji software.

For pDMG cell lines grown as spheroids, HSJD-DIPG-012 were plated in 10% Matrigel (Corning, 354277) in Nunc Lab-Tek Chamber Slide (Thermo Fisher Scientific, 177402). Media and matrigel were supplemented with either 10 ng/mL human recombinant BMP2 or 1 µM LDN-193189 (Selleckchem, S2618). Invasiveness was measured at 48 hr after seeding. Images were captured using an EVOS-7000 (Invitrogen) and analyzed using the 'ROI' feature in the Fiji software.

## Spheroids growth monitoring

To assess the effects of BMP2 on pDMG spheroids, $1×10^4$ cells of HSJD-DIPG-012 or HSJD-DIPG-014 were seeded in 96-well plates (Corning, 7007). Cells were treated at seeding with concentrations of 1, 5, 20, 50, or 200 ng/mL of human recombinant BMP2 (Peprotech, 120–02 C). Media were renewed twice a week, with or without recombinant BMP2. Cell growth was monitored using an EVOS-7000 (Invitrogen) and pictures of control and treated 6 days or 9 days spheres were used to calculate sphere area of HSJD-DIPG-014 and HSJD-DIPG-012, respectively.

## Immunohistochemistry

HSJD-DIPG-012 spheroids were treated when reaching a diameter of 600 µm either with 200 ng/mL of human recombinant BMP2 or with 1 µM of LDN-193189. Spheroids were then fixed in 4% PFA for 1 hr at room temperature, dehydrated, and embedded in paraffin. 8 µm-sections were then deparaffinized, rehydrated, and heated in a citrate buffer (0.01 M; pH 6.0) for 15 min. Sections were then

incubated overnight at 4 °C with the appropriate dilution of anti-Ki67 (1:100, Dako, M7240) in TBS containing Horse Serum (2%). After several washes in TBS-Tween 0.05 %, sections were incubated with the secondary antibody for 1 hr, and then washed again in TBS-Tween 0.05%. Color was developed with 3,3'-diaminobenzidine (DAB, Vector Laboratories, SK-4100) incubation for 1–3 min and with Hematoxylin (Vector Laboratories, H-3401) incubation for 1 min. Images were captured using an EVOS-7000 (Invitrogen) and quantification of Ki67-positive cells was assessed using the 'Cell Positive Detection' QuPath function *Bankhead et al., 2017*, with parameters: 'Optical density sum,' 'thresholdCompartment:' 'Cell: DAB OD mean,' 'thresholdPositive2:' 0.25.

## Statistical analyses

Data are represented as means ± std. Sample size and replicates are stated in the corresponding figure legends. Using GraphPad Prism 9 software and R (4.2.0), the statistical significance between the two groups were determined by one-tailed Mann-Whitney signed-rank tests, apart from the proportion of Ki67 stained cells, which was assessed using Fisher's exact Test. The following symbols used to denote ($<0.05$: *; $<0.01$: **; $<0.001$: ***; $<0.0001$: ****).

## Acknowledgements

We thank the patients and their families who consent to participate in this study, as well as charities that support this work, including the Ligue Nationale contre le Cancer, and the Ligue Départementale contre le Cancer de Haute-Savoie et de l'Ain. Our warmest thanks go to Augustine and its Wonder team ("Wonder Augustine" charity) that support this work since the initiation, to Léonie ("Au Pays de Léonie"), and to "Les Etoiles Filantes", which has funded all the omics analyses. We also thank clinical teams from IHOPe and HFME for their support and contributions, as well as all the facilities from the CLB and CRCL. We are grateful to Séverine Tabone-Eglinger, Loïc Sebileau, and Anne-Sophie Bonne for their help with the management of regulatory procedures. PH and MH received financial support from the Dev2Can LabEx and ITMO Cancer of Aviesan, respectively. CB received financial support from the University of Claude Bernard Lyon 1 in the framework of the "Etoiles 2022" call for projects. S Meyer is supported by a private donation from T. family. The Share4Kids project is supported by INCa and "Enfants Cancers Santé" foundation.

## Additional information

### Funding

| Funder | Grant reference number | Author |
|---|---|---|
| Wonder Augustine | | Marie Castets |
| Nos Etoiles Filantes | | Marie Castets |
| Ligue Contre le Cancer | Enfants, Adolescents et Cancers | Marie Castets |
| Ligue Contre le Cancer - Comités de Haute-Savoie et de l'Ain | | Marie Castets |
| Agence Nationale de la Recherche | Labex DevWeCan 2 | Paul Huchede |
| UCBL | AAP Etoiles | Clément Berthelot |
| INSERM - AVIESAN | FRFT-DOC | Maud Hamadou |

The funders had no role in study design, data collection and interpretation, or the decision to submit the work for publication.

### Author contributions

Paul Huchede, Conceptualization, Formal analysis, Investigation, Methodology, Writing – original draft, Writing – review and editing; Swann Meyer, Conceptualization, Data curation, Formal analysis,

Investigation, Methodology, Writing – review and editing; Clément Berthelot, Maud Hamadou, Adrien Bertrand-Chapel, Conceptualization, Formal analysis, Investigation, Methodology, Writing – review and editing; Andria Rakotomalala, Line Manceau, Julia Tomine, Nicolas Lespinasse, Paul Lewand-owski, Cyril Degletagne, Valéry Attignon, Pascale Gilardi-Hebenstreit, Samuel Meignan, Method-ology; Martine Cordier-Bussat, Laura Broutier, Aurélie Dutour, Jean-Yves Blay, Marion Le Grand, Eddy Pasquier, Conceptualization; Isabelle Rochet, Angel Montero-Carcaboso, Resources, Methodology; Alexandre Vasiljevic, Resources, Formal analysis, Methodology; Pierre Leblond, Conceptualization, Resources, Funding acquisition; Vanessa Ribes, Erika Cosset, Conceptualization, Formal analysis, Methodology, Writing – review and editing; Marie Castets, Conceptualization, Resources, Formal analysis, Supervision, Funding acquisition, Investigation, Methodology, Writing – original draft, Project administration

### Author ORCIDs
Swann Meyer https://orcid.org/0000-0003-3736-4302
Clément Berthelot https://orcid.org/0009-0004-7687-625X
Maud Hamadou http://orcid.org/0009-0008-1460-9032
Adrien Bertrand-Chapel http://orcid.org/0000-0002-0174-4162
Vanessa Ribes https://orcid.org/0000-0001-7016-9192
Marie Castets https://orcid.org/0000-0002-6758-0017

### Ethics
Tissue banking and research were conducted according to national ethics guidelines, after obtaining the written informed consent of patients. This study was approved by the ethical review board of the BRC of the Centre Léon Bérard (no BB-0033-00050, No 2020-02). This BRC quality is certified according to AFNOR NFS96900 (No 2009/35884.2) and ISO 9001 (Certification No 2013/56348.2). Biological material collection and retention activity are declared to the Ministry of Research (DC-2008-99 and AC-2019-3426).

Reviewer #1 (Public Review): https://doi.org/10.7554/eLife.91313.3.sa1
Author response https://doi.org/10.7554/eLife.91313.3.sa2

---

## Additional files

### Supplementary files
• Supplementary file 1. Analysis of the profile of expression of BMP pathway in pDMG. (**a**) List of the BMP target genes used for the heatmaps shown in *Figure 1B* and *Figure 1—figure supplement 1B*. (**b**) Average BMP ligand expression (natural log) per sample in the spatial transcriptomic dataset of DMG samples. (**c**) Normalized expression matrix (vst) of BMP pathway genes from cohort 3, including those shown in *Figure 1—figure supplement 1E*. (**d**) Histone mutation of samples from cohort 3.

• MDAR checklist

### Data availability
Publicly available RNA-seq data of pDMG samples from cohort 1 were obtained from St. Jude Cloud (*McLeod et al., 2021*; https://platform.stjude.cloud/data/diseases/tumor) upon DTA. Cohort 2 of glioma samples from *Mackay et al., 2017* can be accessed on https://pedcbioportal.kidsfirstdrc.org/ with the following dataset id: "phgg_jones_meta_2017". The scRNA-seq cohort of H3.3K27M ACVR1 WT pDMGs is available on GEO (GSE210568) and Zenodo. Share4Kids transcriptomic data from cohort 3 are described in Supplementary file 1c and d. They are available upon request via the Share-4Kids data portal. Users must request access by emailing thomas.diot@lyon.unicancer.fr with a project summary (to comply with the RGPD framework), and access will be automatically granted without any assessment other than checking the ethics of the objectives. They will then have access to a space enabling them to interrogate the data with dedicated analysis tools. RNAseq on glioma cell lines exposed or not to BMP7 are available on GEO (GSE268576). Spatial transcriptomic data on patient samples can be similarly accessed on GEO (GSE268577). The code used to generate the figures has been made publicly available on GitLab under an open source license (copy archived at *Meyer, 2024*).

The following datasets were generated:

| Author(s) | Year | Dataset title | Dataset URL | Database and Identifier |
|---|---|---|---|---|
| Huchedé P, Meyer S, Meignan S, Castets M | 2023 | Transcriptomic modifications induced by BMP depend on H3.3K27M context | https://www.ncbi.nlm.nih.gov/geo/query/acc.cgi?acc=GSE268576 | NCBI Gene Expression Omnibus, GSE268576 |
| Huchedé P, Meyer S, Degletagne C, Vasiljevic A, Castets M | 2023 | Spatial transcriptomic analyses of DIPG samples | https://www.ncbi.nlm.nih.gov/geo/query/acc.cgi?acc=GSE268577 | NCBI Gene Expression Omnibus, GSE268577 |

The following previously published datasets were used:

| Author(s) | Year | Dataset title | Dataset URL | Database and Identifier |
|---|---|---|---|---|
| Jessa S, Hebert S, Kleinman CL | 2022 | HGG-oncohistones processed data | https://www.ncbi.nlm.nih.gov/geo/query/acc.cgi?acc=GSE210568 | NCBI Gene Expression Omnibus, GSE210568 |
| Jessa S, Hébert S, Kleinman CL | 2022 | HGG-oncohistones processed data | https://doi.org/10.5281/zenodo.6929428 | Zenodo, 10.5281/zenodo.6929428 |

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
