## [Editor Report · eLife assessment]

This **valuable** study examines whether the BMP signaling pathway has a role in H3.3K27M DMG tumors, regardless of the presence of ACRVR1 activating mutations. The authors provide **solid** evidence that BMP2/7 synergizes with H3.3K27M to induce a transcriptomic rewiring associated with a quiescent but invasive cell state. Although this work could be further enhanced by the inclusion of additional models, the study overall points to BMP2/7 as a potential target for future therapies in this deadly cancer.

---

## [Referee Report · Reviewer #1 (Public Review)]

Summary:

Mutational analysis of diffuse midline glioma (DMG) found that ACVR1 mutations, which up-regulate BMP signaling pathway are found in most H3.1K27M, but not H3.3K27M DMG cases. In this manuscript, Huchede et al attempted to determine whether the BMP signaling pathway has any role in H3.3K27M DMG tumors. They found that the BMP signaling is activated to a similar level in H3.3K27M DMG cells with wild type ACVR1 compared to ACVR1 DMG cells, likely due to the expression of BMP7 or BMP2. They went on to test whether cells treated with BMP7 or BMP2 treatments affected the gene expression and cell fitness of tumor cells with H3.3K27M mutation. They concluded that BMP2/7 synergizes with H3.3K27M to induce a transcriptomic rewiring associated with a quiescent but invasive cell state. The major issue for this conclusion is that the authors did not use the right models/controls to obtain results to support this conclusion as detailed below. Therefore, in order to strengthen the conclusion, the authors need to address the major concerns below.

Strength:

Address an important question in DMG field.

Major concerns/weakness:

(1) All the results in Fig. 2 utilized two glioma lines SF188 and Res259. The authors should repeat all these experiments in a couple of H3.3K27M DMG lines by deleting H3.3K27M mutation first.

(2) Fig. 3. The experiments of BMP2 treatment should be repeated in another H3.3K27M DMG line using H3.1K27M ACVR1 mutant tumor lines as controls.

Minor concerns

Fig.2A. BMP2 expression increased in H3.3K27M SF188 cells. Therefore, the statement "whereas BMP2 and BMP4 expressions are not significantly modified (Figure 2A and Figure 2-figure supplement A-B)"is not accurate

Comments on revised version:

I had three issues listed above on the initial version. The authors did not address my major concerns of #1 and #2, which are re-listed above.

---

## [Author Response]

The following is the authors’ response to the current reviews.

**Reviewer #1 (Public Review):**

We thank the reviewer for his careful reading, which enabled us to improve the quality of this manuscript. We have addressed some major criticisms, and in particular, we have now included the characterization of the impact of BMP2 on other lines as well as the study of the impact of reversion of the H3.3K27M mutation (Figure 3 - figure supplement 1C-D). This control, judiciously proposed by the reviewer, seems more relevant than using mutant H3.1K27M / ACVR1 lines, given the possibility of BMP2 action via other receptors.

The following is the authors’ response to the original reviews.

**Reviewer #1**
Summary:Mutational analysis of diffuse midline glioma (DMG) found that ACVR1 mutations, which up-regulate the BMP signaling pathway are found in most H3.1K27M, but not H3.3K27M DMG cases. In this manuscript, Huchede et al attempted to determine whether the BMP signaling pathway has any role in H3.3K27M DMG tumors. They found that the BMP signaling is activated to a similar level in H3.3K27M DMG cells with wild-type ACVR1 compared to ACVR1 DMG cells, likely due to the expression of BMP7 or BMP2. They went on to test whether cells treated with BMP7 or BMP2 treatments affected the gene expression and cell fitness of tumor cells with H3.3K27M mutation. They concluded that BMP2/7 synergizes with H3.3K27M to induce a transcriptomic rewiring associated with a quiescent but invasive cell state. The major issue for this conclusion is that the authors did not use the right models/controls to obtain results to support this conclusion as detailed below. Therefore, in order to strengthen the conclusion, the authors need to address the major concerns below.Strength:This paper addresses an important question in the DMG field.Major concerns/weakness:(1) All the results in Fig. 2 utilized two glioma lines SF188 and Res259. The authors should repeat all these experiments in a couple of H3.3K27M DMG lines by deleting the H3.3K27M mutation first.

We thank the referee for his/her comments that have helped us to strengthen our conclusions. Although we were rather interested in studying how the BMP pathway can participate in installing a particular cell state at the time of expression of the K27M mutation, we have now included the characterization of the native H3.3K27M BT245 and SU-DIPGXIII cell lines, and their counterparts in which the mutation was reverted by CRISPRCas9 (Harutyunyan et al., 2019). As shown in Figure 3-figure supplement D, the growth arrest induced by BMP2 seems indeed to be specific of the K27M epigenetic context, which could also be required to settle a positive regulation loop to activate the BMP pathway, as mentioned in the Discussion.

(2) Fig. 3. The experiments of BMP2 treatment should be repeated in other H3.3K27M DMG lines using H3.1K27M ACVR1 mutant tumor lines as controls.

The use of mutant *ACVR1* lines is interesting, but their control status seems questionable, as the addition of BMPs could have a cumulative effect on the effect of the mutation, notably by activating other receptors in the pathway. But we have now included 3 different cell lines (HSJD-DIPG-014, BT245 and SU-DIPGXIII), and observed similar impact of BMP2 with growth arrest as a readout (Figure 3-figure supplement C-D)

Minor concernsFig.2A. BMP2 expression increased in H3.3K27M SF188 cells. Therefore, the statement "whereas BMP2 and BMP4 expressions are not significantly modified (Figure 2A and Figure 2-figure supplement A-B)" is not accurate.

The referee is absolutely right, and we have corrected this statement.

**Reviewer #2 (Public Review):**
The manuscript by Huchede et al investigates the BMP pathway in H3K27M-mutant gliomas carrying or not activating mutations in ALK2 (ACVR1). Their results in cell lines and in datasets acquired from the literature on patient tumors indicate that the BMP signaling pathway is activated at similar levels between ACVR1 wild-type and mutant tumors. The group further identifies BMP2 and BMP7 as possibly the main activators of the pathway in cells. They then show that BMP2 and 7 crosstalk with the H3 mutation and synergize to induce transcriptomic rewiring leading to an invasive cell state.The paper is well-written and easy to follow with a robust experimental plan and datasets supporting the claims. While previous work (acknowledged by the authors) indicated activation of BMP in H3K27M tumors, wild type for the ACVR1 mutation this paper is a nice addition and provides further mechanistic cues as to the importance of the BMP pathway and specific members in these deadly brain cancers. The effect of these BMPs in quiescence and invasion is of particular interest.

We thank the referee for his/her supportive comments.

A few suggestions to clarify the message are provided below 1- In thalamic diffuse midline gliomas, the BMP pathway should not be activated as it is in the pons. The authors should identify thalamic tumors in the datasets they explored and patients-derived cell lines from thalamic tumors available to investigate whether this pathway is active across all H3.3K27M mutants in the brain midline or specifically in tumors from the pons.

The inter-patient variability observed in the level of activation of the BMP pathway may indeed be due, at least in part, to different tumor locations. However, we failed to find this information in the publicly available datasets that we used. We however included this element in the Discussion part.

(2) There are ~20% H3.3K27M tumors that carry an ACVR1 mutation and similar numbers of H3.1K27M that are wild type for this gene. Can the authors identify these outliers in their datasets and assess the activation of BMP2 and 7 or other BMP pathway members in this context?

We have now included the outliers present in our datasets in the legends of Figure 1B and Figure 1-figure supplement B and F. From the few samples available to document these outliers in the cohorts that we used, we have not observed major differences regarding the expression levels of BMP2/7 or BMP pathway members and have discussed the fact that it may result from the establishment in all cases of a feedback loop of activation.

In all this is an interesting paper that provides meaningful data to pursue clinical targeting of the BMP pathway, which would be a nice addition to the field.

We thank the reviewer for his/her supportive comments.